# Risk preference as an outcome of evolutionarily adaptive learning mechanisms: An evolutionary simulation under diverse risky environments

Shogo Homma[1,2,3], Masanori Takezawa[ID][1,4,5]*

**1** Department of Behavioral Science, Graduate School of Humanities and Human Sciences, Hokkaido University, Sapporo, Hokkaido, Japan, **2** Japan Society for the Promotion of Science, Tokyo, Japan, **3** Department of Cognitive and Psychological Sciences, Graduate School of Informatics, Nagoya University, Nagoya, Aichi, Japan, **4** Center for Experimental Research in Social Sciences, Hokkaido University, Sapporo, Hokkaido, Japan, **5** Center for Human Nature, Artificial Intelligence and Neuroscience, Hokkaido University, Sapporo, Hokkaido, Japan

* m.takezawa@let.hokudai.ac.jp

**Data Availability Statement:** All the code for the simulations is available at https://github.com/ShogoHomma/EvolutionRL_Risk.

## Abstract

The optimization of cognitive and learning mechanisms can reveal complicated behavioral phenomena. In this study, we focused on reinforcement learning, which uses different learning rules for positive and negative reward prediction errors. We attempted to relate the evolved learning bias to the complex features of risk preference such as domain-specific behavior manifests and the relatively stable domain-general factor underlying behaviors. The simulations of the evolution of the two learning rates under diverse risky environments showed that the positive learning rate evolved on average to be higher than the negative one, when agents experienced both tasks where risk aversion was more rewarding and risk seeking was more rewarding. This evolution enabled agents to flexibly choose more reward behaviors depending on the task type. The evolved agents also demonstrated behavioral patterns described by the prospect theory. Our simulations captured two aspects of the evolution of risk preference: the domain-specific aspect, behavior acquired through learning in a specific context; and the implicit domain-general aspect, corresponding to the learning rates shaped through evolution to adaptively behave in a wide range of environments. These results imply that our framework of learning under the innate constraint may be useful in understanding the complicated behavioral phenomena.

## Introduction

Behavioral ecology assumes that phenotypes are shaped to maximize long-term fitness through the processes of natural selection, and the underlying genetic architecture and psychological mechanisms that generate these behaviors are often out of the questions. This approach, termed the phenotypic/behavioral gambit [1, 2], has been successful in explaining or predicting animal behavior in a specific environment.

**Funding:** The current study was financially supported by JSPS KAKENHI Grant Number 17H02621, 17H06383, 16H06324 awarded to M. T., 22KJ0056 awarded to S.H, and the foundation for the Fusion of Science and Technology awarded to S.H. The funders had no role in study design, data collection and analysis, decision to publish, or preparation of the manuscript.

**Competing interests:** The authors have declared that no competing interests exist.

However, some behaviors can be better explained by focusing on the optimization of the "cognitive process" involved in choosing the behavior [3]. One example is probability matching, in which the frequency of choosing an option is proportional to the probability of receiving a reward [4]. For example, if option A and B generates a fixed reward with probabilities of 0.7 and 0.3, respectively, most animals tend to choose A 70% of the time and B 30% of the time. The optimization predicts that they should always choose response A to maximize fitness; therefore, probability matching is suboptimal. One possible explanation is that probability matching is a by-product of the evolutionarily optimized reinforcement learning mechanism [5]. The evolution of reinforcement learning can not only explain the suboptimal behavior but complicated behaviors under risk. For example, animals exhibit different risk-sensitive behaviors depending on the risk type [6]. When faced with a risk in the amount of reward (e.g., five rewards for sure vs 10/0 rewards with a probability of 0.5) they show a risk-averse response. When the consequences of a risky behavior are increases in time delay to reward (e.g., a fixed reward is always delivered 15 s later vs. the same reward delivered immediately or 30 s later with a probability of 0.5), they show a risk-seeking response. Buchkremer and Reinhold (2010) [7] show that this behavioral pattern arises from a mechanism that combines reinforcement learning with temporal discounting. They also suggested that the learning and decision mechanism was evolutionarily optimized to maximize a long-term payoff in a typical foraging environment.

Based on this viewpoint, we propose a framework of "learning under innate constraints". Humans and other animals are considered to have innate learning biases. We assumed that behaviors in a specific context are acquired through learning constrained by innate learning biases. The learning mechanism is likely to be tuned such that it allows the agent to acquire adaptive behaviors depending on the context flexibly. Here, the learning bias is defined as a constraint shaped by an evolutionary process. Empirical studies support the notion that learning is limited by innate learning biases (e.g., Garcia effect: [8]). Some researchers have discussed the evolution of biological "preparedness because of its evolutionary advantages" [9, 10]. In this paper, we formulated adaptive learning biases using a reinforcement learning model, and investigated how innate learning biases evolve enabling agents to choose optimal options under a wide range of risky environments.

Explaining decision-making under risk in terms of learning is not new. March (1996) [11] showed that a decision maker who learns from the outcomes of the chosen option showed a behavioral pattern described by the prospect theory [12]. Denrell (2007) [13] theoretically demonstrated that even a risk-neutral learner could behaviorally show risk aversion if the agent could learn only from the outcome sampled from the chosen option. These studies imply that learning would be helpful in analyzing decision-making under risk.

## Overview of reinforcement learning algorithm

Reinforcement learning is a widely used model for learning mechanisms characterized by an algorithm in which an agent learns to choose the optimal behavior in an environment by acquiring rewards through interactions with it [14, 15]. The standard model is termed the temporal difference (TD) learning model [16], the Rescorla-Wagner (RW) model [17, 18], and $Q$-learning model [18]. In the model, in every trial $t$ an agent receives a reward $R_t$, and it updates the value (or the prediction of the outcome when it takes action $a$), $V_t(a)$, as follows:

$$V_{t+1}(a) = V_t(a) + \alpha \cdot \delta_t \tag{1}$$

$$\delta_t = R_t - V_t(a) \tag{2}$$

where $\delta_t$ is reward prediction error (RPE) at trial $t$. RPE is the difference between the actual reward and the prediction. Positive RPE ($\delta_t > 0$) increases the value of the choice and the probability of choosing the option again, whereas negative RPE ($\delta_t < 0$) decreases the value and leads to switching to another option.

The parameter $\alpha$ is called learning rate, which regulates the degree to which the most recent reward experience is used to update the predictions. A higher learning rate significantly changes the prediction by reacting to the latest outcome. The fluctuation in prediction directs an agent to switch choices more frequently. This suggests that learning rate is likely to be a target for the evolutionary selection pressure, particularly in uncertain environments [19].

## Asymmetric learning rates and the evolution

Recent studies proposed a modified version of the model that uses two different learning rates for positive and negative RPE [16, 18, 20, 21]. We refer to this version as the asymmetric reinforcement learning model. Niv et al. (2012) [16] reported that the relative difference between the positive and negative learning rates of a participant was correlated with risky learning behaviors. Individuals who had a larger negative than positive learning rate behaved in a risk-averse manner through learning. Similarly, individuals whose learning rates have opposite relationships learn to engage in risk-seeking behavior.

The asymmetric effects of positive and negative learning experiences are widely known. Several recent studies have shown that humans and some animals overweight positive RPE more than negative RPE in reinforcement learning [22, for a review]. In contrast, it is known that a negative experience leads to faster learning and longer memory retention than a positive one [23, for a review]. Moreover, the evolutionary origins of optimism and pessimism have been widely investigated and discussed in behavioral ecology [24]. Other empirical evidence in cognitive neuroscience shows that this model explains participants' behaviors better than those with a single learning rate [18, 21]. Positive and negative learning rates are also suggested to have neural substrates such as dissociable neural pathways associated with D1 and D2 receptors in the striatum [20]. Therefore, the assumption of two learning rates is a plausible computational learning mechanism.

The evolution of learning bias raises the question of whether learning rates should evolve to perform adaptive behaviors under risk. Some researchers note that learning parameters may vary between animal species [25] and that a bias can evolve as the parameter value constrains the behaviors generated by the algorithm [2, 3]. Previous research focused on the functional aspect of a single learning rate that can control the use of information in a changing environment [26]. This has also been discussed in the field of neuroscience (e.g., [27]). However, how the two learning rates evolve and what patterns of behavior arise through evolution when agents experience multiple tasks in which the risk of the two options varies has not been fully investigated. Cazé and van der Meer (2013) [28] analytically investigated the adaptive relationship between two learning rates in a single two-armed bandit task with binary outcomes. Our study focused on the evolutionary process of the two learning rates when agents experienced multiple risky tasks with continuous outcomes.

## An application of the current framework: The paradox of risk preference

Although our model is conceptual and not designed for directly describing any particular real-world phenomenon, it might be useful for understanding a complex property of risk preference, which is observed in the recent empirical research.

Risk preference (sometimes referred to as risk attitude or sensitivity) is the tendency to respond to risks, the variance in outcome. Risk preference is often implicitly assumed to be a

stable trait within an individual. However, risk preferences have been shown to gradually vary among individuals across domains (e.g., finance, health, and recreation) [29]. Recent research has illustrated the complicated nature of risk preferences: Frey et al. (2017) [30] comprehensively assessed individuals' risk preferences using 39 scales. The scales were categorized into three classes: behavioral, frequency, and propensity measures (i.e., stated preferences). As consistent with the previous findings, the zero-order of correlation coefficients were low within each class ($M = 0.08$–$0.20$) and between classes ($M = 0.03$–$0.13$). Nevertheless, they discovered a general factor of risk preference, $R$, to underlie the measures by using a sophisticated psychometric model. This finding implies that there may be an implicit consistent variance in risk preferences behind the domain-specific measures of risky behaviors. However, it is unclear how different risky behaviors are generated depending on the specific situation while maintaining a relatively consistent domain-general aspect of the behaviors. We refer to this problem as "the paradox of risk preferences." Although the domains discussed in the above research are not precisely incorporated, our conceptual model could provide a potentially useful framework to understand this paradox by formulating both domain-specific adaptations through learning and domain-general learning bias.

## Purpose of this study

We examined how positive and negative learning rates evolve to address a wide range of risky environments and how the evolved learning rates regulate behavioral patterns. To this end, we conducted a computer simulation with an evolutionary process, in which agents with asymmetric reinforcement learning algorithms performed a two-armed bandit task under risky conditions.

The findings of Niv et al. (2012) [16] provide an intuitive prediction of evolutionary results in a single risky environment. Given that the difference between the two learning rates of an agent determines its risky behavior, the value of the negative learning rate should increase or that of the positive learning rate should decrease when risk aversion is more rewarding. Similarly, the opposite direction of evolution should be observed when risk-seeking is more rewarding. First, we examined whether this evolutionary pattern was observed in a risky task, depending on which option was adaptive. Next, we simulated more realistic situations; agents experienced several distinct risky tasks independently during their lifetimes and learned to behave adaptively in response to each environment. This simulation was expected to identify the value of learning rates adapted to a wide range of risky tasks.

As a result of these simulations, we discovered that a negative learning rate had more control over adaptive risky behavior than a positive learning rate (i.e., Fig 3). Surprisingly, the evolution of learning rates through multiple environments led agents to achieve both adaptive risk aversion and risk seeking, depending on the environment they faced. Finally, the population of evolved agents showed behavioral patterns consistent with the prospect theory when the expected values of the two risky options were identical. These results imply that the evolution of learning bias provides a convincing framework for understanding the mechanism behind the paradox of risk preference.

## Results

### Evolution of learning rates in a single-task simulation

Fig 1a illustrates the mean rate of risk aversion in the first and last generations using 35 risk-aversion tasks and 35 risk-seeking tasks. Hereafter, the less risky option is called a safe option. Remember that even the safe option has a variance but is not constant, meaning it is a relative expression. Across the 70 tasks, the population evolved to choose the option with the highest

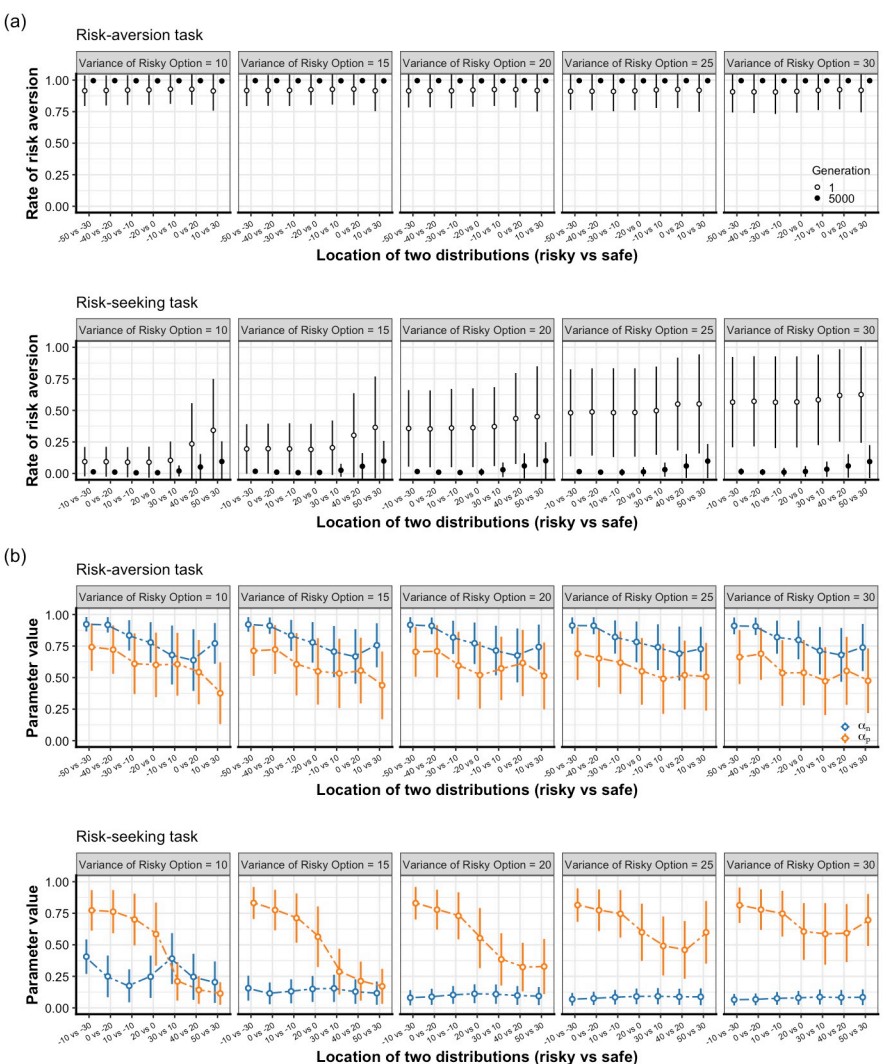

**Fig 1. Results of the single-task simulations.** The horizontal axis represents the location of distributions depicted by $\mu$ (risky option vs safe option). Each panel shows the different standard deviation of the risky option ($\sigma_1$ of normal distribution). The circle is mean value of 10 simulations. The vertical bar is ±1 standard deviation (mean of 10 simulations' SD). (a) Risk aversion rate in 70 single tasks and the comparison between the initial and 5,000th generation. The white and black circles represent the mean risk aversion rate in the initial and last generation, respectively. As a result of evolution, almost all agents successfully learned to choose the option with the higher expected value across 70 tasks. See S1 Fig for the results of the tasks with $D = \pm 10$. (b) Evolutionary result of $\alpha_n$ and $\alpha_p$ in 70 single tasks in 5000th generation. While the relationship $\alpha_n > \alpha_p$ was associated with the risk-aversion tasks, it was reversed ($\alpha_p > \alpha_n$) in the risk-seeking tasks. Note that there were three exceptional risk-seeking tasks that deviated from this pattern: $N(30, 10)$ vs $N(10, 5)$, $N(40, 10)$ vs $N(20, 5)$, and $N(50, 10)$ vs $N(30, 5)$. See S3 Fig for the results of the tasks with $D = \pm 10$.

expected value. In the risk-aversion tasks, agents were able to choose the more rewarding option, even within the first generation. In the risk-seeking tasks, the mean rate of risk aversion ranged between 0.09–0.63 in the first generation. However, most agents were able to choose the higher reward option (risk-seeking option) in the last generation.

Next, we examined the evolution of the learning-rate parameters. Niv et al. (2012) [16] reported that individuals who have larger $\alpha_n$ than $\alpha_p$ tend to exhibit a risk-averse behavior while those who have larger $\alpha_p$ than $\alpha_n$ tend to exhibit a risk-seeking behavior. The

evolutionary dynamics of $\alpha_n$ and $\alpha_p$ reached a steady state within 5,000 generations (S2 Fig). The mean value of $\alpha_n$ evolved to be higher than $\alpha_p$ ($\alpha_n > \alpha_p$) in the risk-aversion tasks while the pattern was reversed ($\alpha_p > \alpha_n$) in risk-seeking tasks (Fig 1b) with some exceptions (the tasks of $N(30, 10)$ vs $N(10, 5)$, $N(40, 10)$ vs $N(20, 5)$, and $N(50, 10)$ vs $N(30, 5)$). The effect size (Cohen's $d$) between $\alpha_n$ and $\alpha_p$ was large in most of the tasks, which shows the mean value of $\alpha_n$ and $\alpha_p$ was sufficiently different (S20 Fig, S4 and S5 Tables). There was an overall tendency for $\alpha_n$ to be more sensitive to natural selection and evolve to take more extreme values than $\alpha_p$. The final mean value of $\alpha_n$ was between 0.63–0.93 in risk-aversion tasks and it quickly reached less than 0.25 in most of the risk-seeking tasks. Conversely, $\alpha_p$ tended to stay close to the initial mean value (i.e., 0.5) in the risk-aversion tasks. In risk-seeking tasks, the evolved $\alpha_p$ was negatively correlated with the location of distributions (this relationship is examined in the next section). Moreover, the standard deviation of the evolved $\alpha_n$ was smaller than that of $\alpha_p$ in the risk-seeking tasks (see S22 Fig and S8 Table for the detailed analysis on the difference of variance between $\alpha_n$ and $\alpha_p$). This suggests that $\alpha_n$ was under stronger selection pressure within our simulation. In addition, because Niv et al. (2012) [16] proposed that the difference between two learning rates is an important factor in predicting the individual risk tendency, we investigated the distribution of $(\alpha_n - \alpha_p)/(\alpha_n + \alpha_p)$, hereafter referred as the Niv index, of the first and last generation (S17 Fig).

We set the inverse temperature $\beta$ as another gene. The evolutionary results are shown in S4 and S5 Figs. The mean $\beta$ value evolved to a higher value (behave in the greedy way) when the task was promoting risk-aversive behavior and when the distributions were in the negative region for the risk-seeking tasks. However, when the distributions were in the positive region during the risk-seeking tasks, $\beta$ evolved to a lower value. In these tasks, if agents with higher $\beta$ initially choose the safe option, even the small positive outcome could greatly increase the probability of choosing the option. Hence agents with lower $\beta$ (those who choose an option in a more random way) were more successful by avoiding getting trapped in the less-rewarding option.

## Evolution of learning rates in a multiple-task simulation

In the previous section, we assumed that all agents faced only a single decision task–either a risk-seeking task or a risk-averse task–throughout their lives. In this section, we assume that agents experience four different tasks in their lives, in which risk-seeking and risk-aversion tasks could be mixed. This multiple-task simulation was conducted to reveal the learning rates that enabled the agents to behave adaptively in a wide range of risky situations. In the simulation, a set of four different 500-trial tasks was randomly selected as an environment, and 1,000 agents played those four tasks independently in their life (the set of four tasks was constant in the simulation).

Fig 2a illustrates the evolutionary dynamics of $\alpha_n$ and $\alpha_p$ in the 100 multiple-task simulations. The value of $\alpha_n$ averaged across simulations evolved to be lower than the initial average value except for the condition where the population only experienced the risk-aversion task. This suggests that the selection pressure to learn to choose a more rewarding risky option decreased the $\alpha_n$ value. Alternatively, the evolved $\alpha_p$ showed larger variability depending on the simulation. This seemingly shows no selection pressure on $\alpha_p$, However, the evolution of $\alpha_p$ was predicted by an exogenous task property, the location of distribution, suggesting that the evolution of $\alpha_p$ was not a random drift. The rate of a negative area of two distributions averaged across four tasks, which partly reflects how negative or positive the two distribution means were located, was positively correlated with the mean value of $\alpha_p$ in the last generation (Fig 2b; the range of Pearson correlation coefficient was from 0.20 when the condition

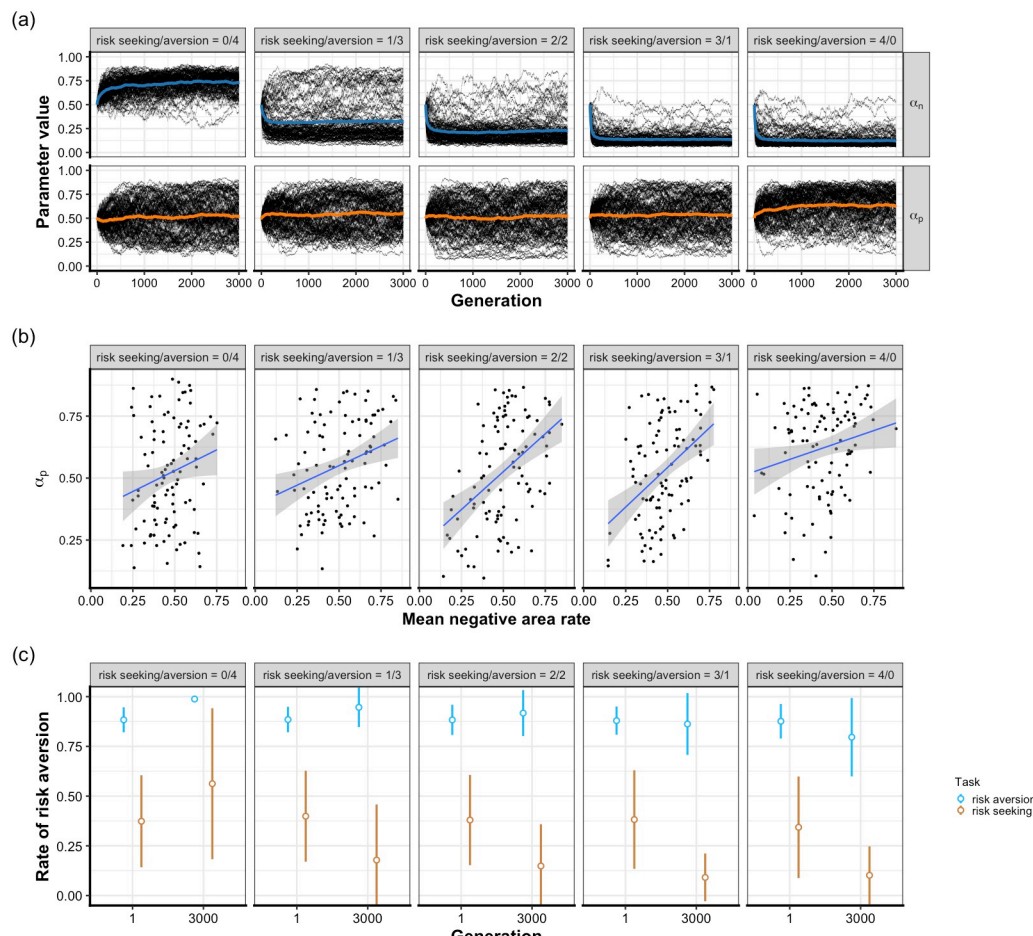

**Fig 2. Results of the multiple-task simulations.** Each column in the panel corresponds to a different condition. (a) Evolutionary dynamics of $\alpha_n$ and $\alpha_p$ in 100 multiple-task simulations. Each black line represents the mean value of a parameter of a simulation. The colored line shows the value averaged across the mean value of simulations. The presence of a risk-seeking task decreased $\alpha_n$. The evolved value of $\alpha_p$ showed greater variability depending on the simulation. (b) The $\alpha_p$ in the last generation was positively correlated with the mean rate of the negative area of two option distributions of the simulation (see S2 Text for the calculation of mean negative area rate). Each point represents the mean value of $\alpha_p$ of a simulation. The blue line is a linear regression line, and the shaded area is 95% confidence interval. (c) Risk aversion rate of the initial and the final generation. The circle and bar correspond to the mean and SD of risk aversion, respectively, across the risk-seeking tasks (dark orange) and the risk-aversion tasks (light blue). The choice data in risk-seeking tasks of 0/4 condition was simulated by randomly sampling a risk-seeking task from the task group and assigning it to the population of the first and last generations for each simulation. The same simulations were done for risk-aversion tasks of 4/0 condition. The mean rate of choosing the more-rewarding option increased in both risk-aversion tasks and risk-seeking tasks except for when the number of the risk-seeking task was three. Overall, behavioral performance was improved through evolution. See S7 for all simulations results.

involved zero risk-seeking task to 0.45 when the condition involved two risk-seeking tasks). The simulations showed that the average relationship $\alpha_p > \alpha_n$ evolved when agents experienced at least one risk-seeking task (see S21 Fig, S6 and S7 Tables for the Cohen's $d$ in the multiple-task simulations). A recent study reported that the bias $\alpha_p > \alpha_n$ contains the tendency to repeat the previously chosen choice [31]. This behavioral tendency is captured using the perseverance parameter. Therefore, the evolved relationship $\alpha_p > \alpha_n$ could disappear by adding the perseverance parameter as a new gene. To address this concern, we simulated a new model that included the perseverance parameter. We discovered that the added parameter did not

influence the results and the same relationship $\alpha_p > \alpha_n$ was replicated (see S3 Text, S12 and S13 Figs). We also investigated the distribution of the evolved Niv index for each condition, which showed the distribution of the Niv index was skewed in the negative direction except 0/4 condition (S18 Fig).

Recall that an evolutionary pattern of decreased $\alpha_n$ was observed in the risk-seeking tasks of the single-task simulation. Thus, it is possible that agents in multiple-task simulations are exclusively adapted to risk-seeking tasks and fail to choose the risk-averse option in the risk-aversion tasks. Surprisingly, the evolved agents were successful in choosing the more rewarding option in both the risk-averse and risk-seeking tasks. Fig 2c shows the rate of risk aversion averaged across the risk-aversion and risk-seeking tasks for each condition. For the condition where the agents experience only risk-aversion tasks (0/4 condition) or risk-seeking tasks (4/0 condition), additional simulations were run to see how the agents in the 0/4 condition would behave in risk-seeking tasks, and the same simulations for the 4/0 condition in the risk-aversion tasks. In risk-aversion tasks, agents were successful in choosing the more rewarding option (i.e., the safe option) even during the first generation. In the risk-seeking tasks, agents performed worse: the mean risk aversion in the risk-seeking tasks was approximately 0.38 in every condition in the first generation. As a result of the evolution, the mean risk aversion in the risk aversion tasks improved slightly or even worsened slightly in the condition of three or four risk-seeking tasks. However, mean risk aversion in the risk-seeking tasks showed consistent great improvement when they experienced risk-seeking tasks in an evolutionary environment. In summary, this result shows that the agents should experience both risk-aversion and risk-seeking tasks so that they can choose the adaptive option in both tasks. This pattern of behavioral improvement was also observed in the learning dynamics within a generation (S8 Fig).

The mean value of $\beta$ basically evolved to be higher in the multiple-task simulations. However, $\beta$ showed more variability as the number of risk-seeking tasks increased (S9 Fig). This evolutionary pattern can be explained by the same logic as that used in the single-task simulation. As the number of risk-seeking tasks increased, the chances that agents got trapped in the less-rewarding option increased; therefore, $\beta$ evolved to reach lower values in some simulations.

## Effect of learning rates on decision-making in a risky situation

Results in the multiple-task simulations with the risk-seeking task showed that $\alpha_n$ evolved to a lower value and the evolved agents adaptively learned to choose the option with a higher expected value regardless of whether it was a risk-aversion or risk-seeking task. To understand the results, we examined the behavior of the agents with various combinations of learning parameter values (Fig 3). In the risk-aversion tasks, agents learned a risk-averse behavior in almost all parameter regions except for where $\alpha_n$ was close to zero (see S23 Fig for the magnified heatmap). Conversely, in the risk-seeking tasks, the choice behavior was greatly altered by the $\alpha_n$ value. The mean value of evolved $\alpha_n$ in the multiple simulations involving both the risk-aversion tasks and risk-seeking tasks was from 0.13 to 0.32, which falls within the parameter range that allows agents to learn the more-rewarding option regardless of the task type. Fig 3 also shows that a slight difference in $\alpha_n$ has a larger impact on the learning of risky behavior, especially in a risk-seeking task, while the difference in $\alpha_p$ is much less influential. Additional logistic regression analysis supported the result in a more quantitative way (S9 Table). This different impact could result in the stronger selection pressure on $\alpha_n$ than that on $\alpha_p$. In addition, the relationship between the Niv index and the choice was investigated (S19 Fig). The positive correlation between the risk aversion and the Niv index was observed as predicted by Niv et al.

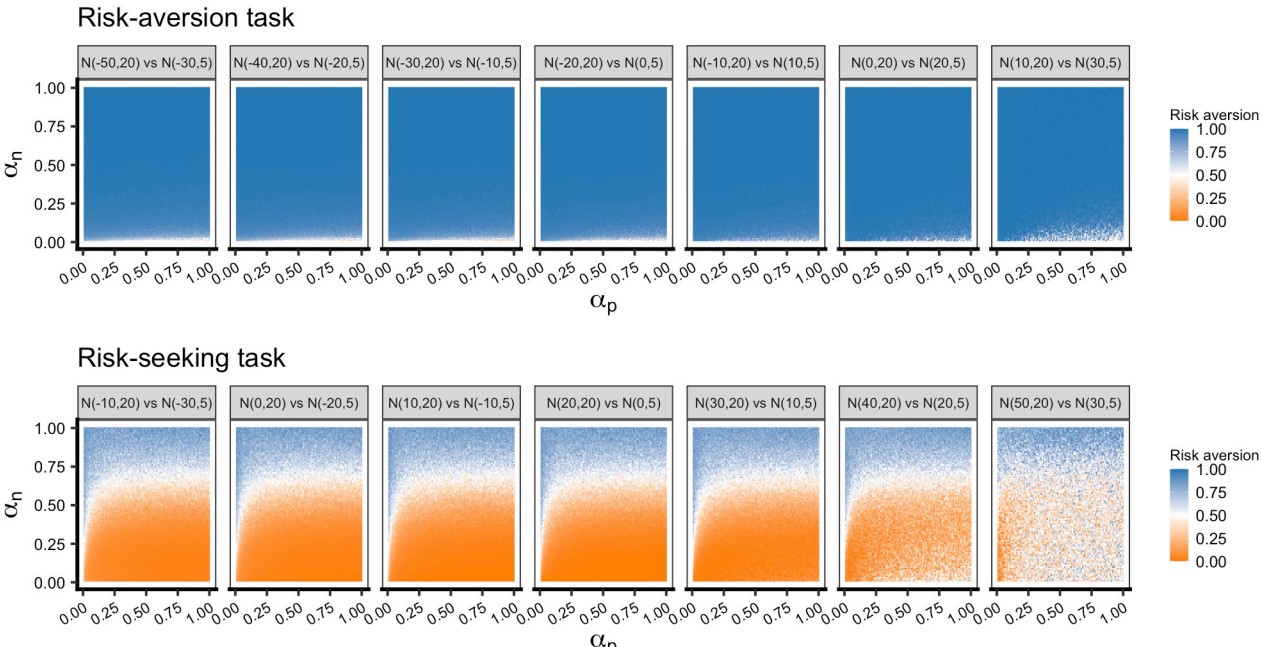

**Fig 3. Effect of $\alpha_p$ and $\alpha_n$ on risky behavior depicted using a heat map.** Each panel is a different location of distributions from negative (left panel) to positive (right panel). Each point represents the mean risk aversion for 500 trials in a specific combination of $\alpha_p$ and $\alpha_n$ in the increments of 0.01. $\beta$ was fixed to 0.25. Dark blue indicates a risk-averse tendency, and dark orange indicates a risk-seeking tendency. In the risk-aversion tasks, agents successfully chose the safe option in a wide range of parameter areas. In the risk-seeking tasks, different $\alpha_n$ greatly affected the risky behavior. See S10 Fig for the results of all tasks and other $\beta$ values.

(2012) [16]. However, a large variance in the choice was observed for the same value of Niv index, which suggests that two learning rates themselves, not the difference, would be better in predicting the choice tendency.

## Risk tendency of the agents who evolved in a multiple-task simulation

In all simulations examined thus far, the expected value of one option was always larger than that of the other, and the optimal option was clearly defined. In human and animal experiments, risk preference is often measured in a task in which the expected values of the two options are identical while the variance is different. In this section, we examine how agents that evolved in a multiple-task simulation behaved when faced with such situations. We extracted the agents from the last generation of 100 multiple-task simulations (a total of 100,000 agents) for each condition. The agents independently learned 500 trials of four different two-armed bandit tasks with equal expected values. The two tasks were gain domain tasks, in which the expected values of the two options were both positive ($N(20, 20)$ vs. $N(20, 5)$, and $N(10, 20)$ vs. $N(10, 5)$). The remaining two tasks were loss domain tasks, in which the expected values were both negative ($N(-10, 20)$ vs. $N(-10, 5)$ and $N(-20, 20)$ vs. $N(-20, 5)$).

Interestingly, agents who experienced at least one risk-seeking task in an evolutionary environment showed a behavioral pattern described by the prospect theory. More agents exhibited risk-averse tendencies when the task was in the gain domain. The histogram of the risk aversion rate exhibits a bimodal distribution; the largest number of agents exhibit almost complete risk-averse behavior, followed by almost complete risk-seeking behavior (left two panels in Fig 4a). However, in the loss domain, the peak of the distribution decreased, indicating that

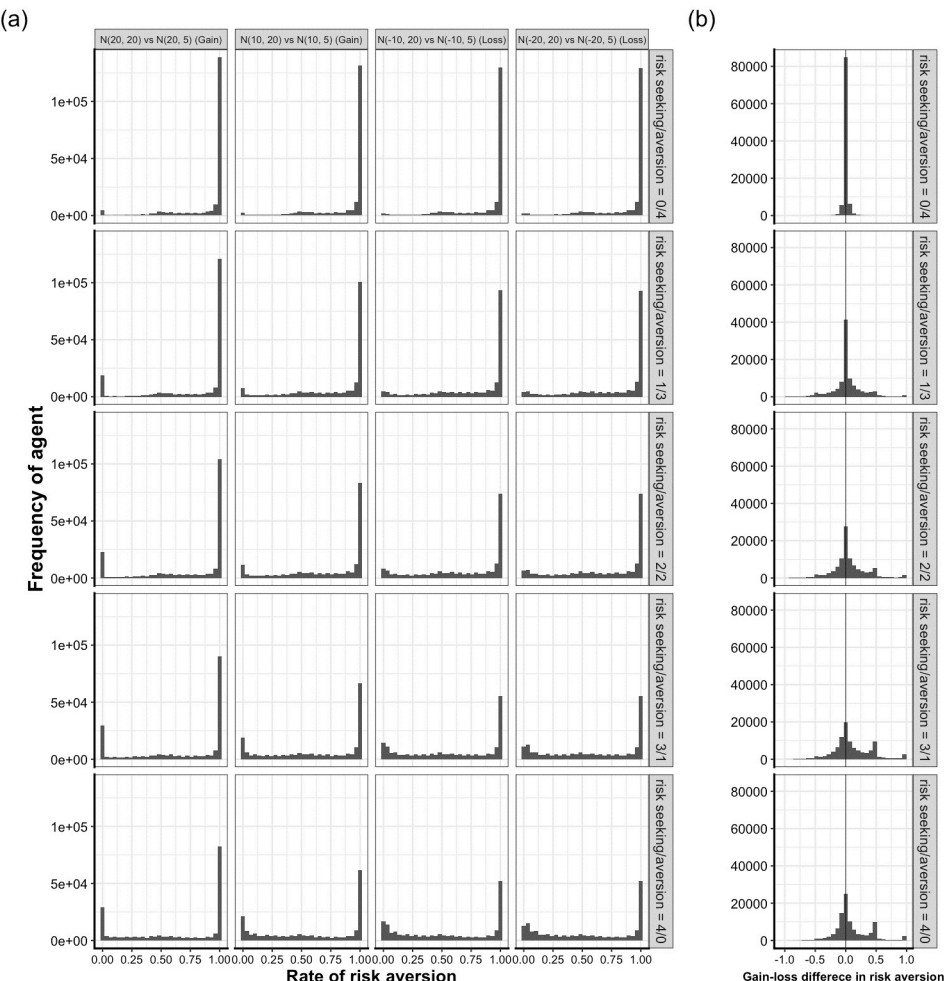

**Fig 4. Risk-tendency of the agents who evolved in a multiple-task simulation.** (a) Frequency of risk aversion rate when evolved agents played the tasks where the expected value of the two options was the same. The rate of risk aversion was calculated from the last 100-trial choice to exclude the behaviors of early trials, which are considered to be not fully learned behaviors. Each column panel represents the four different tasks from the gain (left two panels) to the loss domain (right two panels). In the gain domain tasks, more agents behaved in a risk-averse way. In the loss domain task, the number of agents who show risk-seeking increased. (b) Frequency of the difference in risk aversion rate of gain minus loss domain. The mean rate of risk aversion across two gain domain tasks and that across two loss domain tasks were calculated for each agent. If the value of the difference is positive, the agent showed more risk aversion in the gain domain. More agents changed their behavior to a more risk-seeking attitude for the loss domain task.

the number of agents who showed risk-seeking behavior increased (right two panels in Fig 4a). We calculated the change in the frequency of risk aversion between the gain and loss domains for each agent (Fig 4b). The rates of agents who changed their behavior to a more risk-seeking approach in the loss domain (the behavioral change between gain and loss was more than zero) were 45.1%, 51.8%, and 55.4% when the number of risk-seeking tasks was one, two, and three, respectively, in the evolutionary environment. This means that agents who evolved in an environment involving both risk-averse and risk-seeking tasks behaved in a risk-averse manner in the gain domain and in a more risk-seeking manner in the loss domain, which is the typical behavioral pattern described by the prospect theory. This pattern was less evident when the agents in the first generation performed the same task (S11 Fig).

## Discussion

Learning systems enable organisms to adapt to a wide range of environments. Moreover, learning can be optimized through evolution to acquire adaptive behavior more efficiently. We focused on the reinforcement learning algorithm and examined how the learning rates to the positive and negative RPE ($\alpha_p$ and $\alpha_n$, respectively) are tuned through the evolution when it experiences diverse risky situations.

In the single-task simulation, while $\alpha_n$ consistently increased and led to the parameter relationship $\alpha_n > \alpha_p$ when the safe option was more rewarding (risk-aversion task), $\alpha_n$ decreased to result in $\alpha_p > \alpha_n$ when the risky option was more rewarding (risk-seeking task). In the multiple-task simulation, where agents could experience both risk-aversion and risk-seeking tasks in their lifetime, $\alpha_n$ decreased through evolution, and the relationship $\alpha_p > \alpha_n$ evolved on average. Interestingly, the evolved agents in the multiple-task simulation successfully adapted to both risk-averse and risk-seeking tasks. Furthermore, when the evolved population faced a task in which the expected values of the two options were the same, they showed the behavioral pattern described by the prospect theory: risk-averse behavior in the gain domain and risk-seeking behavior in the loss domain.

The results that the evolved agents are successful in choosing the adaptive option in both risk-aversion and risk-seeking tasks need to be discussed in relation to the previous research [13, 32, 33]. Denrell (2007) [13] analyzed a single learning rate model and also showed that, regardless of the value of the learning rate, agents learn to choose a risk-averse option in a risk-aversion task, while learning the risky option in a risky task is possible only with particular learning rate values. Functional aspects of reinforcement learning were also investigated by Denrell (2005) under a single risky environment [32] and by Toyokawa and Gaissmaier (2022) under social conditions [33]. These previous studies mainly used a normal distribution for the risky option, which was similar to our study. Although our study is related to these previous studies, and especially to Denrell's (2007) findings, our study is also different from them. First, we analyzed the two learning rate model and found that $\alpha_n$ is more decisive in controlling the adaptive choice. Second, we further analyzed the multiple-task environments and validated the generality of the results obtained in a single-task environment. Third, we found that the agents evolved in multiple task environments exhibited behavioral patterns consistent with the prospect theory. Our study thus accumulated novel contributions to the previous studies.

We focused on the model with two learning rates from the perspective of neural implementation. We also simulated the reinforcement learning with the single learning rate ($\alpha$). As a result of the evolution of $\alpha$ and $\beta$, the agents showed similar behavioral performance in the multiple-task simulations as the model with the two learning rates and the same risk tendency when they played the task with the same expected values (see S4 Text, S14–S16 Figs). This suggests that a model with single learning rate may be equivalent to one with two learning rates at the functional level in our simulations. One of the implications is that the tasks we used might be too simple, so agents did not show a difference in behavioral performance. Another implication is that the two-learning rate systems may not be outcomes of functional adaptations, but outcomes of a phylogenetic factor or physiological constraint.

We define the paradoxical aspect of risk preference as a complicated property that implicitly maintains a stable individual difference (the general factor of risk preference) while dynamically changing within an individual depending on the context. Our results provide a conceptual framework that captures the fundamental aspects of risk preference. The general factor corresponds to an evolved learning bias $\alpha_p > \alpha_n$, and the varying aspect is interpreted as the behavior which is acquired through learning in a specific context. As for the individual difference in the learning bias, $\alpha_n$ almost consistently evolved to a lower value; therefore, small

variations in $\alpha_n$ were observed. In contrast, the evolution of $\alpha_p$ showed more variability depending on the location of the two distributions. Hence, the main source of variation in a learning bias can be $\alpha_p$. Our simulation suggests that the variation can arise from whether the individual experiences a greater gain or loss environment. The domain specificity of risk preference is adaptation to a specific risky situation through learning. This perspective is partially supported by the simulation results, in which the expected values of the two options are the same. The population behaved as described by the prospect theory, one of the most successful descriptive models of decision-making under risk. A previous theoretical study that examined the evolution of risk-averse tendencies in humans assumed that risk-averse/risk-seeking behavior (or preference) is genetically determined as a strategy [34]. Our study proposes that a complicated behavioral pattern can be captured by assuming that a phenotype is acquired through learning.

Our finding that the behavioral pattern described by the prospect theory was observed in the evolved population seems intriguing from the description-experience gap perspective [35]. Our simulation assumes that individuals continuously update their predictions based on their experiences. This type of decision-making corresponds to "decision by experience." Conversely, the prospect theory aims to describe human choice behavior in "decision by description" settings, where the decision-maker chooses an option, explicitly given information about the outcome and probability of the posed options by symbols or written sentences. Probability has been culturally constructed and has spread throughout the modern period [36]. In contrast, decision-making based on experience is commonly observed in animals. Moreover, the mechanisms of value calculation behind prospect theory and reinforcement learning differs. In the prospect theory, the prospective valuation of outcomes is involved in loss aversion, whereas the retrospective valuation of outcomes (prediction error) works in reinforcement learning [22]. It is not our argument that our model replaces the prospect theory. It is an important future question why and how similar behavioral patterns emerged from different mechanisms underlying the decisions by description and experience.

Our simulation showed that $\alpha_n$ is consistently influenced by natural selection unlike $\alpha_p$, and the value of $\alpha_n$ influenced the agents' behavior more strongly than $\alpha_p$. The smaller variance of the evolved $\alpha_n$ in the single-task simulation also suggests that it was under stronger selection pressure. This indicates that $\alpha_p$ and $\alpha_n$ have an asymmetric effect on behaviors. The $\alpha_p$ (or $\alpha_n$) controls the degree to which the agent repeats (or switches from) the last choice when the outcome exceeds (or worse than) the prediction. It is sufficient that agents do not react to a negative RPE from a risky option and persist in it in a risk-seeking task, and that they learn and switch from it in a risk-aversion task. Previous research analyzing the similar model with a single learning rate suggested that a learning rate which exceeds a certain threshold produces a risk-averse tendency, even though the risky option is more rewarding [37]. In our model, the similar phenomenon appears to arise from a sensitive reaction to a negative RPE. This effect should be accelerated by increasing $\alpha_n$ in a risk-aversion task and should be hindered by decreasing $\alpha_n$ in a risk-seeking task. Moreover, since a risky option yields higher RPE by definition, how agents react to the outcome of the risky option matters, not that of the safe option. Hence, the learning rate can have a larger impact on behavior when agents continue choosing the risky option (in a risk-seeking task).

The role of $\alpha_n$ is more complicated. In a risk-seeking task, it seems adaptive to have a higher $\alpha_p$ and keep the value of the risky option higher. However, higher $\alpha_p$ can also increase the chance that agents get trapped in a safe option when the expected distributions values are positive, given that the initial value of $V$ is zero. Hence, when the distributions are positively located, it may be more adaptive to have a lower $\alpha_p$. In contrast, when the distributions are negatively located, it is more adaptive to have a higher $\alpha_p$ because there is a small chance to get

trapped. However, whether the proposed mechanism can be verified quantitatively remains unclear. It will be necessary to examine the asymmetric impact of $\alpha_p$ and $\alpha_n$ on learning in a theoretical and quantitative way.

As a result of the adaptation to diverse risky situations, the relationship $\alpha_p > \alpha_n$ evolved and $\alpha_p$ showed a larger variability. An emerging question is how empirically valid the distributions of $\alpha_p$ and $\alpha_n$ are. A recent review reports that the average relationship $\alpha_p > \alpha_n$ is consistently observed in humans and some animals in experimental research, and that it is more advantageous in a wide range of task structures than a model with the single learning rate [22]. Previous research investigating the advantage of $\alpha_p > \alpha_n$ focused on binary outcomes [38–40] while the current research focused on continuous outcomes. Our study contributes to the literature by demonstrating that the $\alpha_p > \alpha_n$ is an outcome of the evolution of reinforcement learning under diverse risky environments, and that the evolved agents exhibit the risk-averse tendency (or more specifically behavior described by prospect theory) under the task with both options having the same expected value.

Each two-armed bandit task in the multiple-task simulation may be interpreted as corresponding to different domains of risky situations (e.g., finance, health, and recreation). The simulations assumed that all tasks were non-volatile, that is, the expected value and variance of the distribution were constant across trials and generations. It is more common to assume that some task parameters, especially the expected value, change continuously or suddenly during learning or between generations. Adjusting the learning rate is important for choosing adaptive behavior in volatile environments in mathematical biology [26] and neuroscience [27], suggesting that volatility is an ecological factor that influences learning rate evolution. A recent unpublished study showed that $\alpha_p > \alpha_n$ is advantageous in a volatile environment [38]. However, the study did not consider the effects of the different risks between options. Future research should examine the adaptive learning bias under risk and volatility as well as whether it can capture the decision-making observed in reality. In addition, the accessibility to information of foregone payoffs needs to be considered as another important feature of an environment. The learning of risk-averse behavior depends on whether the agent has access to the forgone (unchosen option's) payoff or not [13]. The information about the forgone payoff provides agents with opportunities to improve the value of a devaluated risky option. Because the function of the evolved small $\alpha_n$ is considered to inhibit the devaluation, the access to the forgone payoff would weaken the selection pressure on $\alpha_n$. However, it is natural to assume that reinforcement learners need to be tuned to an environment where foregone payoff information is probabilistically available. Like the multiple-task environment in our simulations, such partial accessibility to the foregone payoff information may recover the selection pressure on $\alpha_n$.

In conclusion, our results suggest that the complicated aspect of risk preference can be captured by learning bias in reinforcement learning as an innate constraint shaped by an evolutionary process. This also sheds light on the idea of optimization at the level of cognition and learning, not on the behavior itself. The perspective that behaviors emerge from a cognitive mechanism optimized through evolution will help us understand complicated or seemingly irrational behaviors.

## Model

In the following simulations, asexual $N$ agents played a two-armed bandit task over $T$ trials independently. At the end of each generation, the agent produces offspring. This process is repeated for $G$ generations in each simulation.

## Two-armed bandit task

In the two-armed bandit task, an agent chooses one of two options. Payoffs are randomly sampled from a normal distribution $N(\mu, \sigma)$, whose mean ($\mu$) and standard deviation ($\sigma$) are fixed for each option and across trials. The standard deviation of option 2 is small, and that of option 1 is always larger ($\sigma_1 > \sigma_2$). Hereafter, we refer to Option 1 as the risky option and Option 2 as the safe option. Note that all the agents in a population perform the same task.

## Algorithm of learning

The agent behavior was calculated using a reinforcement learning algorithm with an asymmetric learning rate [16],

$$V_{t+1}(a) = V_t(a) + \begin{cases} \alpha_p \cdot \delta_t \text{ if } \delta_t \geq 0 \\ \alpha_n \cdot \delta_t \text{ if } \delta_t < 0 \end{cases} \tag{3}$$

$$\delta_t = R_t - V_t(a),$$

where $V_t(a)$ is the value (expected reward) of the action of choosing option $a$ at trial $t$. $V_1(a)$, the values of the action in the first trial are set to zero; $\alpha_p, \alpha_n \in [0, 1]$ are positive and negative learning rate, respectively. They are a weighting parameter on positive RPE ($\delta_t \geq 0$) and negative RPE ($\delta_t < 0$). The value of the unchosen option $\bar{a}$ is not updated ($V_{t+1}(\bar{a}) = V_t(\bar{a})$). Agents choose an action with the probability calculated using the softmax rule,

$$p_t(a = 1) = \frac{1}{1 + \exp\left(-\beta(V_t(1) - V_t(2))\right)} \tag{4}$$

The parameter $\beta$ is inverse temperature and the smaller value leads to more exploration (i.e., choose more randomly) and the larger value leads to exploitation (i.e., more likely to choose the action with larger $V(a)$).

It is possible to consider the reinforcement learning with the single learning rate as a candidate model. We also simulated the model in the multiple-task simulation. The model with the evolved $\alpha$ and $\beta$ showed almost the same behavioral performance as that with two learning rates. See the discussion section for the details and results implications.

## Evolution of parameters

Three parameters, $\alpha_p$, $\alpha_n$, and $\beta$, were assumed to be genetically transmitted. At the beginning of the simulation, the parameter values of each agent were randomly sampled from a uniform distribution (Uniform[0, 1] for $\alpha_p$ and $\alpha_n$; Uniform[0, 0.5] for $\beta$). Here, we set the upper bound of $\beta$ for the convenience, see S1 Text for the detail.

At the end of each generation, an agent produced offspring with a probability proportional to the fitness, the mean of the payoffs across all trials, and a baseline fitness of 20. If the minimal value of the population fitness was negative, it was adjusted to have a minimal value 0.1 by adding the absolute value and 0.1 to the population. Note that fitness is an arithmetic mean; therefore, bet-hedging is not considered in our simulation. The mutation was modeled by adding Gaussian noise each gene. The Gaussian noise was randomly sampled from $N(0, 0.01)$ for $\alpha_p$ and $\alpha_n$, and $N(0, 0.005)$ for $\beta$. The distribution was truncated such that the gene value was within the range after a mutation.

## Simulations setup

This research mainly conducted three types of simulations: single-task simulations, multiple-task simulations, and simulations for measuring agents' risk tendencies. In the single-task simulation, 10,000 agents performed a single task for 500 trials within a generation. We systematically change the location of the expected value of the two distributions and the relative magnitude of the variance (risk) of the risky option. Let $D$ be the difference between the means of the two options ($\mu_1 - \mu_2$). When $D$ is positive ($\mu_1 > \mu_2$), or the risky option is more rewarding, it is called a risk-seeking task. When $D$ is negative ($\mu_1 < \mu_2$), it is called a risk-aversion task. Positively located distributions (options) indicate that the initial prediction of agents is lower than the true value of options (i.e., an agent's pessimistic expectation for options), whereas negatively located distributions represent the opposite (i.e., an agent's optimistic expectation for options). In total, 140 tasks were simulated (see S1 Text for the detail of the task structures). We mainly report the case of $D = \pm 20$ (70 tasks) because almost the same results were obtained from the case with $D = \pm 10$ (See the S1–S3 Figs). We ran ten independent simulations for each task setting and reported the average results.

In the multiple-task simulation, a population of agents performed four different tasks for 500 trials within each generation. The value of the options ($V_t(a)$) in each task was updated independently (i.e., the experience in a task was not generalized to other tasks). The initial values were set to zero across all options in each task ($V_0(a) = 0$). Because $\alpha_p$, $\alpha_n$, and $\beta$ were assumed to be genetically transmitted, agents updated action values using the same parameter values across the tasks. Fitness was calculated as the arithmetic mean of all acquired payoffs across all four tasks. Moreover, we manipulated the number of risk-seeking tasks that the agents experienced in their lives as a simulation condition, from zero to four. To cover a wide range of task structures, we ran 100 multiple-task simulations for a condition, in which four tasks were randomly determined in a specific protocol. See S2 Text and S6 Fig for the detailed information on the protocol and the generated tasks.

The final simulations were conducted to measure the risk tendencies of the evolved agents. We simulated and assessed the frequency of risk-averse choices of agents who evolved in multiple-task simulations under tasks in which the expected values of the two options were the same but the variance differed. The agents independently played 500-trial two-armed bandit tasks, where the expected values of the two options were both positive (gain-domain task) and negative (loss-domain task). Because the initial value of the two options was zero, the gain (or loss) domain represented a situation in which the initial reference point was lower (or higher) than the actual value of the options.

## Supporting information

**S1 Text. Detail of the single-task simulation.**
(PDF)

**S2 Text. Detail of the multiple-task simulation.**
(PDF)

**S3 Text. Reinforcement learning involving the perseverance parameter.**
(PDF)

**S4 Text. Reinforcement learning using a single learning rate.**
(PDF)

**S1 Fig. Risk aversion rate in 70 single tasks with D = ±10 in the first and last generation.**
The horizontal axis represents the location of distributions depicted by $\mu$ (risky option vs safe

option). Each panel corresponds to the different risks of the risky option. The white and black circle indicates the mean rate of risk aversion in the first and last generation, respectively (the averaged result of 10 simulations with the same task parameter setting). The vertical bar is ±1 standard deviation (mean of 10 simulations' SD). As reported in the main text (tasks with $D = \pm20$), agents evolved to choose the more rewarding option in the last generation.
(PDF)

**S2 Fig. Comprehensive display of evolutionary dynamics of $\alpha_n$ and $\alpha_p$ in 140 single-task simulations.** The solid line and colored area ($\alpha_n$ is blue and $\alpha_P$ is orange) are the mean value and SD, respectively, which were averaged over 10 simulations conducted with the same parameter setting. The column indicates the risk of the risky option ($\sigma_1$). The row indicates the normal distribution of the safe option. Each panel corresponds to a single task. The task distribution is depicted by "risky option vs safe option" inside a panel. In the main text, we only report the mean value of parameters in the last generation. This figure shows that the mean value of learning rates becomes almost steady within 5,000 generations in almost all of tasks.
(PDF)

**S3 Fig. Evolutionary result of $\alpha_n$ and $\alpha_p$ in 70 single tasks with D = ±10.** The horizontal axis represents the location of distributions. Each panel corresponds to the different risks of risky option ($\sigma_1$). The white circle indicates the mean value in the last generation (the averaged result of 10 simulations with the same task parameter setting). The vertical bar is ±1 SD (mean of 10 simulations' SD). As reported in the main text (tasks with $D = \pm20$), the parameter relationship $\alpha_n > \alpha_p$ was consistently observed for the risk-aversion tasks, and the reversed relationship $\alpha_p > \alpha_n$ was observed for the risk-seeking tasks.
(PDF)

**S4 Fig. Comprehensive display of evolutionary dynamics of $\beta$ in 140 single-task simulations.** The solid line and colored area are the mean value and SD, respectively, which were averaged over 10 simulations conducted with the same parameter setting. The column indicates the risk of the risky option ($\sigma_1$). The row indicates the normal distribution of the safe option. Each panel corresponds to a single task. The task distribution is depicted by "risky option vs safe option" inside a panel.
(PDF)

**S5 Fig. Evolutionary result of $\beta$ in 140 single tasks.** The white circle represents the mean value in the last generation (mean of 10 simulations with the same parameter setting). The vertical bar is ±1 standard deviation (mean of 10 simulations' SD). In the positively located risk-seeking tasks, the lower mean value of $\beta$ evolved to avoid being trapped in the less-rewarding option. In the other tasks, the mean $\beta$ value evolved to reach higher values and allowed agents to behave in a greedy manner. The mean $\beta$ value evolved to a higher value (consistently more than 0.35) in the risk-aversion task and when the distributions were in the negative region for the risk-seeking tasks. As the expected values of the two options did not change during learning, it seems reasonable that the agents tended to choose the more valuable option in a greedy manner. However, when the distributions were in the positive region during the risk-seeking tasks, the mean $\beta$ evolved to a lower value. In these tasks, if agents with higher $\beta$ initially choose the safe option, even the small positive outcome could greatly increase the probability of choosing the safe option. This leads agents to get trapped in the less-rewarding safe option. Hence agents with lower $\beta$ (those who choose an option in a more random way) were more successful by avoiding safe option in the risk-seeking task.
(PDF)

**S6 Fig. Histogram of (a) the effect size and (b) negative area rate in a task group of a multiple-task simulation.** Each panel represents a simulation condition.
(PDF)

**S7 Fig. Comprehensive display of the risk aversion rate of all tasks in multiple-task simulations.** The horizontal axis indicates a task in a simulation depicted by the simulation number (from 1 to 100)–task number (from 1 to 4). The white circle and black circle represent the mean rate of risk aversion in the first and last generation, respectively. The vertical bar is ±1 SD.
(PDF)

**S8 Fig. Learning dynamics of the population in the multiple-task simulations as well as comparison of the first generation with the last generation.** The solid line represents the mean rate of risk aversion. The colored area shows ±1 SD. The top (bottom) panel corresponds to the first (last) generation. The mean rate of choosing the more rewarding option increased in both the risk-aversion and risk-seeking tasks for the final generation.
(PDF)

**S9 Fig. Evolutionary dynamics of $\beta$ in 100 multiple-task simulations.** Each black line represents the mean value of the parameter of a simulation. The colored line shows the averaged value of the mean value of simulations. As the number of risk-seeking tasks increased, the potential to be trapped in the less-rewarding option increased, which led $\beta$ evolving to lower in some simulations. The mean $\beta$ increased when the number of risk-seeking task was small, and it showed more variability as the number of risk-seeking tasks increased. This evolutionary pattern can be explained by the same logic as that used in the single-task simulation. When the number of risk-seeking tasks was small, the mean value of $\beta$ evolved to be higher (led agents to behave in the greedy way). However, as it increased, the chances that agents got trapped in the less-rewarding safe option also increased; therefore, $\beta$ evolved to be lower in some simulations (led agents to behave in the random way).
(PDF)

**S10 Fig. Comprehensive display of the effects of $\alpha_p$ and $\alpha_n$ on risky behavior in the 70 single-task simulations when $\beta$ was fixed to 0.1, 0.25, or 0.4.** The column indicates the risk of the risky option ($\sigma_1$). The row indicates the normal distribution of the safe option. Each panel corresponds to a single task. The task distribution is depicted by "risky option vs safe option" inside a panel. The star (*) following the task distribution indicates that the panel was shown in Fig 3 in the main text.
(PDF)

**S11 Fig. Risk tendency of evolved agents in the tasks where the expected values are the same.** (a) Frequency of agents' risk aversion rate in the initial generation of multiple-task simulations when the expected value of two options was the same. The rate of risk aversion was calculated from the last 100-trial choice of 500 trials. Because the initial genes were randomly assigned from a uniform distribution, the histograms were almost identical across simulation conditions. Furthermore, regardless of the gain or loss domain, a peak was observed for complete risk aversion. (b) The frequency of the difference in risk aversion rate between the gain domain and loss domain in the initial agents. A high peak in the difference was observed at approximately zero; thus, indicating that numerous agents slightly changed their degree of risk aversion. The number of agents who increased their risk aversion in the gain domain was small compared with those in the last generation.
(PDF)

**S12 Fig. Evolutionary dynamics of parameters in the hybrid model.** The evolution of (a) $\alpha_n$ and $\alpha_p$, (b) $\beta$, and (c) $\varphi$. (d) A positive correlation was observed between the evolved $\alpha_p$ and the mean rate of the negative area of the two option distributions of the simulation. The almost same result was replicated as the asymmetric reinforcement learning model: that is, when agents experience at least one risk-seeking task, the value of $\alpha_n$ decreased and on average the relationship $\alpha_n > \alpha_p$ was observed. The inverse temperature ($\beta$) showed a larger variability as the number of risk-seeking task increased. The mean value of $\varphi$ evolved from 1.19 to 1.83 except for the condition where agents only experienced the risk-aversion task (the mean value was 3.05).
(PDF)

**S13 Fig. Behavioral performance of the hybrid model.** (a) Comparison of the risk aversion rate between the first and final generations. Circles and vertical bars represent the mean and SD of risk aversion in the risk-seeking (dark orange) and risk-aversion tasks (light blue), respectively. (b) Comparison of average learning dynamics throughout trials between the first generation (top panel) and last generation (bottom panel). The solid line represents the mean risk aversion rate. Colored area shows ±1 SD. As with the asymmetric reinforcement learning model, the mean value of risk aversion improved in both risk-seeking and risk-averse tasks as a result of evolution.
(PDF)

**S14 Fig. Evolutionary dynamics of parameters of the single learning rate model.** The evolution of (a) $\alpha$ and (b) $\beta$. The averaged value of $\alpha$ decreased except the condition where agents only experience the risk-aversion task, which was similar to the evolution of $\alpha_n$ within the asymmetric reinforcement learning model. The averaged $\beta$ value increased regardless of the simulation condition.
(PDF)

**S15 Fig. Behavioral performance of the reinforcement learning model with the single learning rate.** (a) Comparison of risk aversion rate between the first and final generation. Circles and vertical bars represent the mean and SD of risk aversion in the risk-seeking task (dark orange) and the risk-aversion tasks (light blue). (b) Comparison of the averaged learning dynamics through trials between the first generation (top panel) and last generation (bottom panel). The solid line represents the mean rate of risk aversion. The colored area shows ±1 SD. Although the single learning rate model has fewer parameters, it exhibits almost the same performance as the asymmetric reinforcement learning model.
(PDF)

**S16 Fig. Frequency of risk aversion rate of the single learning rate model in the tasks where the expected value of the two options was the same.** Rate of risk aversion was calculated from the last 100-trial choice of 500 trials. Histogram of the risk aversion rate of agents (a) in the first generation and (c) in the last generation. The difference of risk aversion rate between gain and loss domain (b) in the first generation and (d) in the last generation. If the difference is positive, the agent showed more risk aversion in the gain domain. Most agents in the first generation showed complete risk aversion regardless of the task domain and simulation condition, and the second-highest mode was observed at an intermediate level of risk aversion. As a result of this evolution, more agents tended to exhibit complete risk aversion when they experienced more risk-aversion tasks (e.g., the condition in which the number of risk-seeking tasks was zero or one). In the other evolutionary condition, the frequency of complete risk aversion decreased in the loss domain. Similar to the asymmetric reinforcement learning model, the evolved agents showed more risk aversion in the gain domain than in the loss domain (or

more risk seeking in the loss domain). The rate of behavioral change between gain and loss was more than zero at 43.6%, 50.3%, and 52.5% when the number of risk-seeking tasks was one, two, and three in the evolutionary environment, respectively (see S3 Table).
(PDF)

**S17 Fig. Histogram of Niv index in the single-task simulation.** The horizontal axis represents the Niv index $(\alpha_n - \alpha_p)/(\alpha_n + \alpha_p)$ calculated for each agent. The tasks are the same as Fig 1 in the main text. The column indicates the SD of the risky option $(\sigma_1)$. The row indicates the location of two distributions depicted by $\mu$ (risky vs safe option). Red and blue color correspond to the histogram of the first and last generation, respectively. In risk-aversion tasks, the histogram skewed to the positive value. In risk-seeking tasks, the histogram skewed to the negative value with three exceptions.
(PDF)

**S18 Fig. Histogram of Niv index in the multiple-task simulation.** The horizontal axis represents the Niv index $(\alpha_n - \alpha_p)/(\alpha_n + \alpha_p)$ calculated for each agent. Each column in the panel corresponds to a different condition. Red and blue color correspond to the histogram of the first and last generation, respectively. The histogram skewed to the negative value when agents experience a risk-seeking task.
(PDF)

**S19 Fig. Relationship between Niv index and risk aversion.** The scatter plot was created from the data of Fig 3 of the main text. Each point represents an agent. The dashed line is the fitted linear regression line. Pearson correlation coefficient is shown in the bottom. The positive correlation was observed between Niv index and the proportion of choosing the safe option. However, a large variance in the choice was found even though the value of Niv index was the same.
(PDF)

**S20 Fig. Histogram of Cohen's $d$ in the single-task simulation.** Cohen's $d$ for the evolved $\alpha_p$ and $\alpha_n$ was calculated for each task by $d = \left(M_{\alpha_p} - M_{\alpha_n}\right)/\sigma_{\text{pooled}}$ and $\sigma_{\text{pooled}} = \sqrt{\left(SD^2_{\alpha_p} + SD^2_{\alpha_n}\right)/2}$, where $M$ and $SD$ is the mean and standard deviation of an parameter in a task, respectively. The positive value of $d$ indicates that the mean value of $\alpha_p$ is larger than that of $\alpha_n$ while the negative value of $d$ indicates the opposite relationship. See S4 Table for the statistics of Cohen's $d$ and S5 Table for the detailed value of each task.
(PDF)

**S21 Fig. Histogram of the Cohen's $d$ in the multiple-task simulation.** Each column in the panel corresponds to a different condition. The Cohen's $d$ of the evolved $\alpha_p$ and $\alpha_n$ was calculated for each simulation. See the caption of S20 Fig for the calculation. The histogram shows that the most of the Cohen's $d$ is distributed in a positive range when agents experience a risk-seeking task. See S6 Table for the statistics of Cohen's $d$ and S7 Table for the summary of effect size.
(PDF)

**S22 Fig. Histogram of SD ratio in the single-task simulation.** The ratio of standard deviation of the evolved $\alpha_p$ to that of the evolved $\alpha_n$ was calculated for each task. Most of the SD ratio was greater than one, suggesting that the SD of the evolved $\alpha_p$ was sufficiently larger than that of the evolved $\alpha_n$.
(PDF)

**S23 Fig. Magnified heatmap of risk-aversion task.** The heatmap of risk-aversion task of Fig 3. was magnified by dividing the y-axis $(\alpha_n)$ by 0.1. The figure illustrates that risk-neutral

or risk-seeking tendency can be found around the area where $\alpha_n$ is close to zero, but not found in other area.
(PDF)

**S1 Table. Task configuration in the multiple-task simulation.**
(PDF)

**S2 Table. Frequency of agents who showed more risk aversion in the gain domain than in the loss domain.**
(PDF)

**S3 Table. Frequency of agents who showed more risk aversion in the gain domain than in the loss domain when the model used a single learning rate.**
(PDF)

**S4 Table. Summary of statistics of Cohen's *d* in the single-task simulation.**
(PDF)

**S5 Table. Detailed value of Cohen's *d* and effect size in the single-task simulation.**
(PDF)

**S6 Table. Summary of statistics of Cohen's *d* in the multiple-task simulation.**
(PDF)

**S7 Table. Frequency of effect size in the multiple-task simulation.**
(PDF)

**S8 Table. Summary of statistics of SD ratio in the single-task simulation.**
(PDF)

**S9 Table. Summary of estimated parameters of logistic regression analysis.**
(PDF)

## Acknowledgments

We are thankful to Wataru Toyokawa and Hideki Ohira for their helpful comments on the early draft.

## Author Contributions

**Conceptualization:** Shogo Homma, Masanori Takezawa.

**Formal analysis:** Shogo Homma, Masanori Takezawa.

**Funding acquisition:** Shogo Homma, Masanori Takezawa.

**Investigation:** Shogo Homma, Masanori Takezawa.

**Methodology:** Shogo Homma, Masanori Takezawa.

**Project administration:** Masanori Takezawa.

**Validation:** Shogo Homma, Masanori Takezawa.

**Visualization:** Shogo Homma.

**Writing – original draft:** Shogo Homma, Masanori Takezawa.

**Writing – review & editing:** Shogo Homma, Masanori Takezawa.

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
