## [Decision Letter · Decision Letter 0]

29 Nov 2023

PONE-D-23-31441Risk preference as an outcome of evolutionarily adaptive learning mechanisms: an evolutionary simulation under diverse risky environmentsPLOS ONE

Dear Dr. Takezawa,

Thank you for submitting your manuscript to PLOS ONE. After careful consideration, we feel that it has merit but does not fully meet PLOS ONE’s publication criteria as it currently stands. Therefore, we invite you to submit a revised version of the manuscript that addresses the points raised during the review process.

The manuscript presents engaging insights into agent-based models within evolutionary contexts, particularly focusing on the evolution of risk aversion and risk-seeking behaviors through reinforcement learning models. However, both reviewers have identified substantial conceptual and methodological concerns that warrant attention. Specifically, issues have been highlighted regarding the interplay between learning rate bias and risk-taking behaviors. Reviewer 1 has questioned the conceptual originality of the findings, referencing aspects such as the 'hot stove effect' and prospect theory. Reviewer 2 has elaborated on methodological challenges, especially in relation to the treatment of parameters within the simulation framework. In light of these critiques, it is imperative that the authors provide comprehensive responses to these reviewer comments to enhance the rigor and clarity of their study.

We look forward to receiving your revised manuscript.

Kind regards,

Rei Akaishi

Academic Editor

PLOS ONE

2. We are unable to open your Supporting Information file [__MACOSX]. Please kindly revise as necessary and re-upload.

Additional Editor Comments:

The manuscript presents engaging insights into agent-based models within evolutionary contexts, particularly focusing on the evolution of risk aversion and risk-seeking behaviors through reinforcement learning models. However, both reviewers have identified substantial conceptual and methodological concerns that warrant attention. Specifically, issues have been highlighted regarding the interplay between learning rate bias and risk-taking behaviors. Reviewer 1 has questioned the conceptual originality of the findings, referencing aspects such as the 'hot stove effect' and prospect theory. Reviewer 2 has elaborated on methodological challenges, especially in relation to the treatment of parameters within the simulation framework. In light of these critiques, it is imperative that the authors provide comprehensive responses to these reviewer comments to enhance the rigor and clarity of their study.

Reviewers' comments:

Reviewer's Responses to Questions

**Comments to the Author**

1. Is the manuscript technically sound, and do the data support the conclusions?

Reviewer #1: Yes

Reviewer #2: Partly

2. Has the statistical analysis been performed appropriately and rigorously? 

Reviewer #1: N/A

Reviewer #2: No

3. Have the authors made all data underlying the findings in their manuscript fully available?

Reviewer #1: Yes

Reviewer #2: Yes

4. Is the manuscript presented in an intelligible fashion and written in standard English?

Reviewer #1: Yes

Reviewer #2: Yes

5. Review Comments to the Author

Reviewer #1: Dear Editor,

The authors report the results from a series of agent-based model simulations where they studied the evolution of reinforcement learning parameters under multiple risky two-armed bandit tasks. It had been well-known that a reinforcement learning often shows risk aversion in some ranges of parameter combinations even if there is a positive risk premium in choosing risky options. Such a potentially suboptimal risk-aversion would not always emerge, and reinforcement learning could evolve to be risk-seeking if natural selection pressures act on a longer-term payoff obtained a single risky bandit task. However, if agents' fitnesses were determined by multiple different bandit tasks, it was unclear how reinforcement learning would evolve and what type of risk-taking biases would emerge through natural selection.

To address this, the authors assumed that agents had to perform four different bandit tasks whose risk profile was systematically varied. The authors found that agents evolved a sort of flexible payoff-maximising bias, that is, showing risk aversion when the risk premium was negative (i.e. the risk aversion tasks), while resulting in risk seeking when the risk premium was positive (i.e. the risk seeking tasks). For someone who has been studying reinforcement learning and risk taking behaviour, this result itself did not seem to be so surprising and it was just consistent with what we knew.

What was interesting though was that there were two learning rate parameters assumed, one was for positive RPEs and another for negative RPEs, which were evolving separately. It was also very nice that the authors had thoroughly compared the main results with both a simpler version of RL with a single learning rate and a more complex candidate model with a choice trace parameter (called a hybrid model). Then the authors found that it was mostly a negative learning rate that shaped agents' risk taking biases, which makes totally sense because avoiding underestimation of risky option is key to prevent the hot stove effect.

Overall, although the findings were not so surprising to someone in the decision science, the evolution of learning behaviour and how it relates to domain general risk taking biases would be of interest of wider audience including behavioural and evolutionary ecologists. The study is technically very sound and the authors present it in a concise way. However, before accepting this manuscript for publication, there are several key points that I believe the authors should address in a revision.

Major points:

(1) Relate more to the previous works on the hot stove effect

The authors should not overstate the novelty of their results, and should pay fair credits to previous works. For instance, the current manuscript reads that "Surprisingly, the evolved agents were successful in choosing the more rewarding option in both the risk-averse and risk-seeking tasks"; however, it was not surprising at all, I am afraid. The simple RL model considered here cannot become risk-seeking in any range of parameter combinations when the risk premium is negative (See for example Eq. (2) in Denrell (2007)). In other words, RLs always learn to become risk aversive irrespective of the task profile unless the risk premium was large enough. This implies in principle that RL agents who have successfully evolved to be risk seeking under the risk-seeking tasks, still should always be risk aversive under the risk-aversive tasks regardless of whatever the learning rates and the inverse temperature they evolved. In general, I would appreciate if the authors can point out how their findings can relate to the well-known hot stove effect, highlighting what is truly news in the literature and what is a replication of the previous literature (which is also nice things to see).

(2) The connection to the gain/loss domain was misleading

The authors investigated the behaviour of evolved agents by varying the positions of agents' initial q-values compared to the tasks' average reward value. When agents started the task with an optimistic prior belief, they tended to be more explorative at the beginning, which made it easier to overcome the hot stove effect when the risk premium was positive. On the other hand, when they had an pessimistic prior, they were prone to get stuck in the safer option. This result makes totally a sense, but has nothing to do with the gain/loss domain, let alone with the prospect theory that was about the decisions from description but not from experience. RL agents do not care about the "domain" of the payoff because what they learn is a difference between their current q-value and payoff just obtained (that is RPEs). They could have integrated the PT by assuming that learning were based on agents' subjective utilities rather than payoffs, that is, utility = f(payoff). But I guess that that direction of modelling would not be what they aimed to do here.

Having said that, the point that the evolved agents' behaviour was consistent with what PT predicted was interesting. This might imply that human risk taking behaviour in the gain and loss domains may not always be a result of the biased subjective utility and probability as PT claimed. Instead, the same behavioural pattern could have been generated by simple RL mechanisms that do not care about whether it is the gain or loss domain.

So in sum, I disagree with what the authors discussed in lines 432 - 436. What we observed in this manuscript has no direct relations to PT. Instead, in a revision, the authors could discuss what it would mean that two different mechanisms (that is, PT and RL) happen to show a similar pattern. If the authors still believe that what was observed in their results could be relevant to PT, more detailed discussion would be needed. I think this might also be related to the description and experience gap:

Hertwig, R., & Erev, I. (2009). The description–experience gap in risky choice. Trends in cognitive sciences, 13(12), 517-523.

(3) What about the evolved variations in alphas?

The evolved variation in alpha_p was larger than that of alpha_n. It would be nice if the authors can explain or discuss what this pattern might imply.

Minor comments:

(4) Lines 110 - 112: As for the evolution of the Rescorla-Wagner type of learning rate, the authors could cite this:

P.C. Trimmer, J.M. McNamara, A.I. Houston, J.A.R. Marshall. Does natural selection favour the Rescorla-Wagner rule? Journal of Theoretical Biology, 302 (2012), pp. 39-52, 10.1016/j.jtbi.2012.02.014

(5) Also, though they are not evolutionary studies, the following studies have investigated functional aspects of reinforcement learning mechanisms under different kinds of risks and social environments. In my view, one of key factors of the current study was that they focused on multiple tasks simultaneously. The citations below were analysing the function of RL under only a single environment. So I guess citing them could make it easier to highlight the uniqueness of the authors' current study using multiple bandits.

Denrell, J. (2005). Why most people disapprove of me: experience sampling in impression formation. Psychological review, 112(4), 951.

Toyokawa, W., & Gaissmaier, W. (2022). Conformist social learning leads to self-organised prevention against adverse bias in risky decision making. eLife, 11, e75308.

(6) On Fig. 2(C): To back up their argument that having at least one risk-seeking environment would promote the evolution of the flexible risk taking (that is, risk seeking only under the positive risk premium), they should also check how agents who had evolved under the risk aversion only condition (that is, the "0/4" condition) would behave in a risk seeking task. For example, the authors could show such data on Fig. 2C too (for instance, running a new risk-seeking task with "0/4" agents and show their risk aversion bias at the 3000th generation of the "0/4" panel). They might not be able to show risk seeking if their evolved environment was "0/4". The same could be done for the "4/0" condition too.

I am looking forward to the revision.

All the best

Reviewer #2: The study “Risk preference as an outcome of evolutionarily adaptive learning mechanisms: an evolutionary simulation under diverse risky environments” presents the evolution of positive and negative learning rates in various risky contexts. The authors conducted a series of simulations with reinforcement learning models and found that, over generations, the negative learning rate became higher than the positive learning rate when a safe option (i.e., with lower variance) was associated with a higher mean outcome. Conversely, the negative learning rate became smaller than the positive learning rate when the risky option (i.e., with higher variance) was associated with a higher mean outcome. These patterns were consistent not only within single risky contexts but also in environments with multiple risky contexts. The authors also established an association between learning rates and risk attitude through simulations with different combinations of learning parameter values. Furthermore, the evolved learning rates were applied to tasks where the mean of outcome distributions was controlled for risky and non-risky options. This revealed a heightened tendency for risk aversion in the gain domain compared to the loss domain, particularly when the evolved learning rates originated from tasks featuring at least one risk-seeking context. Overall, the authors conducted a systematic investigation into the evolution of learning rates and their relationship with risk attitudes. However, I have three major points (and a few minor ones) that could enhance the clarity of the conclusions.

Major points:

1.Conceptual: It remains unclear if risk preference is related to learning bias (e.g., the difference between positive and negative learning rate) or learning rate (either positive or negative) itself. If the authors are interested in the learning bias (as mentioned in the introduction and conclusion), then I would expect more statistical analysis on the difference between positive and negative learning rates (see also the next two points). For example, the correlation between learning bias and level of risk aversion. So far, the results and interpretation mainly focus on the role of each learning rate (e.g., how does each learning rate evolve; “a negative learning rate had more control over adaptive risky behavior than a positive learning rate” (Line 189)), which makes me wonder how much evidence supports the conclusion “the complicated aspect of risk preference can be captured by learning bias in reinforcement learning as an innate constraint shaped by an evolutionary process.” (Line 502-504)

2.Analysis: One of the authors’ summaries is that “ is consistently influenced by natural selection unlike , and the value of influenced the agents’ behavior more strongly than ”. If this is the case, I have three follow-up questions:

a. Why did the authors focus on a positive learning rate rather than a negative learning rate (or learning bias: – ) in Figure 2b?

b. Although the choice pattern is dominated by the negative learning rate (αn), the variance of choice is similar to the variability of positive learning rate (αp) especially in the risk-seeking task in Figure1. Therefore, I wonder, is it possible that behavior is influenced by two learning rates. For example, the index of risk aversion is related to the difference between p and (Niv et al., 2012). The result will provide additional (and more straightforward) insight into “how evolved learning rates regulate behavioral patterns” (Line 174).

3.More statistical results are required to support the arguments. According to the visual inspection in Figure 1 and 3, the arguments “a negative learning rate had more control over adaptive risky behavior than a positive learning rate” and “the evolved showed larger variability depending on the simulation” are true for the risk-seeking task only. With the same figures, however, it is unclear whether the same arguments also apply to the risk-aversion task. Specifically, in the risk-aversion task

a. whether the difference between αn and αp is significant? If yes, then the current conclusion holds. Otherwise, the conclusion might be restricted to the risk-seeking task.

b. whether the variability of αp is significantly larger than the variability of αn?

c. whether the impact of αn is larger than the impact of αp on the risk attitude? Again, this is not clear from the Figure 3.

Minor points:

1.Could the authors elaborate on the relationship between risk sensitivity and probability matching more in the introduction section? It is difficult for me to understand why risk sensitivity could be used to explain probability matching.

2.Line410-418. Please elucidate the unique role of the two learning rates when compared to a single learning rate model in shaping risk attitudes. Specifically, if a single learning rate model can generate equivalent choice patterns to what the two-learning-rate model predicts, it is essential to clarify the added value or unique capabilities of the learning bias that the single learning rate model cannot provide. Importantly, the explanation is helpful for readers to understand the relationship between risk preference and learning bias (which is not available in the single-learning-rate model).

3.Figure3 suggests that risk attitudes are influenced by context, even when learning components, such as the positive learning rate (p) and positive learning rate (n), are 0 or close to 0. To gain a deeper insight into the significance of the learning mechanisms in shaping risk attitudes, I would like to know why the differences in risk attitudes between risk-aversion and risk-seeking tasks persist despite values are not (or slowly) updated.

4.Line 79-82. Please clarify the definition of optimal options in the sentence. If the optimal option refers to the option associated with a higher anticipated outcome (i.e., the non-risky option in the risk-aversion task and the risky option in the risk-seeking task), then the evolution of fitness should be observed. Is that the case? On the other hand, I might misunderstand how the authors computed the probability of producing offspring. Could the authors clarify why the baseline fitness is 20 rather than the mean of the payoff across agents? I am wondering if the probability can be larger than 1 if the mean of the payoff is larger than 20 (e.g., in the context of 30 vs 50 or -30 vs -50).

5.The current study establishes a connection between learning rates and risk preferences, particularly emphasizing that individuals tend to exhibit greater risk aversion when n > p, aligning with prior findings (Niv et al., 2012). However, from an evolutionary standpoint, it raises the question of how the authors interpret earlier research that suggests humans have a tendency toward positivity bias (i.e., p > n) and an inclination towards risk aversion.

6.Learning strategy can be context-dependent. Therefore, it would be great if the authors could provide further insights into the extent to which the findings from the current study can be extended to alternative scenarios. For instance, can these results be applied to situations where agents have access to both chosen and unchosen outcomes, or when agents begin making assumptions about the foregone outcome (Ben-Artzi et al., 2023; Ting et al., 2022; Bavard et al., 2021).

7.Figures:

a. The font size in each figure could be enlarged.

b. What does each dot represent in Figure 2b?

c. The description “… agents learned a risk-averse behavior in almost all parameter regions except for where was close to zero.” in Line 326 is not visible in Figure 3.

8.Suggestions for wording:

a. Figure: rate of risk averse -> probability of choosing non-risky option or p(choosing safe option)

b. Figure: risk of the risky option -> variance of the risky option

c. Line 343: agents successfully chose the risk-averse behavior -> agents successfully chose the risky option

References:

1. Ben-Artzi, I., Kessler, Y., Nicenboim, B., & Shahar, N. (2023). Computational mechanisms underlying latent value updating of unchosen actions. Science Advances, 9(42), eadi2704.

2. Ting, C. C., Palminteri, S., Lebreton, M., & Engelmann, J. B. (2022). The elusive effects of incidental anxiety on reinforcement-learning. Journal of Experimental Psychology: Learning, Memory, and Cognition, 48(5), 619.

3. Bavard, S., & Palminteri, S. (2021). Why unchosen options linger in our minds. Communications Biology, 4(1), 1271.

4. Niv, Y., Edlund, J. A., Dayan, P., & O'Doherty, J. P. (2012). Neural prediction errors reveal a risk-sensitive reinforcement-learning process in the human brain. Journal of Neuroscience, 32(2), 551-562.

6. PLOS authors have the option to publish the peer review history of their article (what does this mean?). If published, this will include your full peer review and any attached files.

Reviewer #1: No

Reviewer #2: No

---

## [Author Response · Author response to Decision Letter 0]

29 May 2024

We included all the comments to the editor and the reviewers in the response letter.

---

## [Decision Letter · Decision Letter 1]

16 Jul 2024

Risk preference as an outcome of evolutionarily adaptive learning mechanisms: an evolutionary simulation under diverse risky environments

PONE-D-23-31441R1

Dear Dr. Takezawa,

We’re pleased to inform you that your manuscript has been judged scientifically suitable for publication and will be formally accepted for publication once it meets all outstanding technical requirements.

Kind regards,

Rei Akaishi

Academic Editor

PLOS ONE

Additional Editor Comments (optional):

The paper is basically accepted. Reviewer 2 made some comments. Please follow the suggestions of the reviewer 2.

Reviewers' comments:

Reviewer's Responses to Questions

**Comments to the Author**

1. If the authors have adequately addressed your comments raised in a previous round of review and you feel that this manuscript is now acceptable for publication, you may indicate that here to bypass the “Comments to the Author” section, enter your conflict of interest statement in the “Confidential to Editor” section, and submit your "Accept" recommendation.

Reviewer #2: All comments have been addressed

2. Is the manuscript technically sound, and do the data support the conclusions?

Reviewer #2: Yes

3. Has the statistical analysis been performed appropriately and rigorously? 

Reviewer #2: Yes

4. Have the authors made all data underlying the findings in their manuscript fully available?

Reviewer #2: (No Response)

5. Is the manuscript presented in an intelligible fashion and written in standard English?

Reviewer #2: Yes

6. Review Comments to the Author

Reviewer #2: The revision successfully clarifies the primary concern regarding the of positive and negative learning rates in shaping risk attitudes. I have a few suggestions for the authors to further minimize potential confusion around some terms used in the paper. Notably, even without implementing these suggestions, I believe this work could be published in PLOS ONE.

1. It would be beneficial for the authors to add a sentence in the Results section emphasizing that the "safe" option is not entirely free of risk. This clarification would help readers understand that the safe option still involves a certain level of uncertainty (albeit lower than the risky option). Specifically, when discussing the Results section, adding this context will make it easier to interpret comparisons such as N(30, 10) vs. N(10, 5) (i.e., Line 201).

2. The authors have included a few sentences in the Discussion section to highlight the similarities and differences between their work and previous studies. I would encourage the authors to also emphasize the similarities and differences in the simulation settings between their study and prior research. For instance, they could cite papers that used the same definition of risky alternatives and the applied changes in the variance of risky options.

7. PLOS authors have the option to publish the peer review history of their article (what does this mean?). If published, this will include your full peer review and any attached files.

Reviewer #2: No

---

## [Editor Report · Acceptance letter]

24 Jul 2024

PONE-D-23-31441R1 

PLOS ONE

Dear Dr. Takezawa, 

I'm pleased to inform you that your manuscript has been deemed suitable for publication in PLOS ONE. Congratulations! Your manuscript is now being handed over to our production team.

Kind regards, 

on behalf of

Dr. Rei Akaishi 

Academic Editor

PLOS ONE